# Identification of New EGFR Inhibitors by Structure-Based Virtual Screening and Biological Evaluation

**DOI:** 10.3390/ijms25031887

**Published:** 2024-02-04

**Authors:** Shuyi Wang, Xiaotian Xu, Chuxin Pan, Qian Guo, Qinlan Li, Shanhe Wan, Zhonghuang Li, Jiajie Zhang, Xiaoyun Wu

**Affiliations:** Guangdong Provincial Key Laboratory of New Drug Screening, School of Pharmaceutical Sciences, Southern Medical University, Guangzhou 510515, Chinaqguo610@163.com (Q.G.); lzhuang@smu.edu.cn (Z.L.);

**Keywords:** epidermal growth factor receptor (EGFR), virtual screening, molecular dynamics

## Abstract

Epidermal growth factor receptor (EGFR) inhibitors have been used in clinical for the treatment of non-small-cell lung cancer for years. However, the emergence of drug resistance continues to be a major problem. To identify potential inhibitors, molecular docking-based virtual screening was conducted on ChemDiv and Enamine commercial databases using the Glide program. After multi-step VS and visual inspection, a total of 23 compounds with novel and varied structures were selected, and the predicted ADMET properties were within the satisfactory range. Further molecular dynamics simulations revealed that the reprehensive compound ZINC49691377 formed a stable complex with the allosteric pocket of EGFR and exhibited conserved hydrogen bond interactions with Lys 745 and Asp855 of EGFR over the course of simulation. All compounds were further tested in experiments. Among them, the most promising hit ZINC49691377 demonstrated excellent anti-proliferation activity against H1975 and PC-9 cells, while showing no significant anti-proliferation activity against A549 cells. Meanwhile, apoptosis analysis indicated that the compound ZINC49691377 can effectively induce apoptosis of H1975 and PC-9 cells in a dose-dependent manner, while having no significant effect on the apoptosis of A549 cells. The results indicate that ZINC49691377 exhibits good selectivity. Based on virtual screening and bioassays, ZINC4961377 can be considered as an excellent starting point for the development of new EGFR inhibitors.

## 1. Introduction

Non-small-cell lung cancer (NSCLC), the most frequently reported subtype of all lung cancer cases, makes up 80–85% of the disease. The receptor tyrosine kinase superfamily member epidermal growth factor receptor (EGFR) promotes the development of NSCLC [1], which is frequently caused by activating mutations in *EGFR* kinase. The most common mutations are deletion mutation in exon 19 and L858R point mutations changing from leucine to arginine at position 858 in the EGFR kinase. First generation EGFR inhibitors, exemplified by the US Food and Drug Administration (FDA)-approved gefitinib and erlotinib, are effective for advanced NSCLC [2,3]. Despite the effectiveness of these inhibitors in clinical trials, drug resistance arises due to various mechanisms, including T790M mutation [4]. The gatekeeper T790M mutation, which resulted from the replacement of threonine with methionine at position 790 in the EGFR ATP-binding pocket, was discovered in almost half the patients treated with these drugs. The T790M mutation improves ATP affinity thereby decreasing the effectiveness of reversible ATP-competitive inhibitors, leading to drug resistance [5]. Second- and third-generation EGFR inhibitors, exemplified by Afatinib, Dacomitinib, and Osimertinib, covalently target the T790M mutant by reacting with Cys 797 residue adjacent to the EGFR ATP site [6]. However, resistance developed through different mechanisms, and C797S mutation serves as a primary mechanism for resistance. The irreversible covalent EGFR inhibitors become clinically ineffective, because they are unable to form a covalent bond with EGFR after the change of Cys797 to a less reactive serine residue [7,8,9].

All currently FDA-approved EGFR kinase inhibitors are ATP competitive inhibitors, specifically targeting the highly conserved ATP binding site [10], which highlights the need for new inhibitors with different mechanisms of action [11]. Recently, mutant selectivity inhibitors EAI001 (Figure 1) were identified [12], which showed high potency for EGFR^L858R/T790M^ with IC_50_ value of 24 nM. Following further optimization, EAI045 achieved more potent activity against EGFR^L858R/T790M^ with IC_50_ values of 3 nM and exhibited about 1000-fold high selectivity against the WT EGFR. EAI001 binds to the allosteric binding pocket that is created by the displacement of the C-helix rather than the ATP-binding pocket, as confirmed by the co-crystal structure of EAI001 bound to EGFR. The released crystal structure provides the possibility of conducting structure-based virtual screening (SBVS) to discover allosteric inhibitors with unique chemical scaffolds. Molecular docking has been recognized as the most popular method for SBVS [13]. It can predict the binding conformations of the ligands and rank the ligands according to scoring functions. In this study, we aim to discover potential EGFR inhibitors with diverse scaffolds through high throughput virtual screening and biological evaluation. The workflow of the current study is represented in Figure 2. Molecular docking was conducted on ChemDiv and Enamine commercial databases. After multi-step VS and visual evaluation, a total of 23 structurally unique and diverse compounds were chosen and the predicted ADMET properties are within the satisfactory range. Molecular dynamics simulations revealed that the reprehensive compound ZINC49691377 formed a stable complex with EGFR. All selected compounds were further subjected to biological evaluation. Among them, ZINC49691377 showed good anti-proliferation activity against H1975 and PC-9 cells, while showing no significant anti-proliferation activity against A549 cells. Apoptosis analysis indicated that ZINC49691377 can effectively induce apoptosis of H1975 and PC-9 cells in a dose-dependent manner, while having no significant effect on apoptosis of A549 cells. The results indicated that ZINC4961377 with a new scaffold can be considered as a good point for further optimization.

## 2. Results and Discussions

### 2.1. Molecular Docking Protocol Assessment

In this research, to explore the possibility of discovering potential EGFR inhibitors, a virtual screening was carried out on commercial databases using the co-crystal structure of EAI001 bound to EGFR (PDB ID: 5D41) [12], which has a resolution of 2.31 Å. All the programs used were from Schrödinger Suite package 2017-1, and the widely used Glide program [14,15] was used for docking and virtual screening. First, in order to validate the reliability of three docking protocol, the native EAI001 was re-docked into the binding site using HTVS (high-throughput virtual screening), SP (standard precision) and XP (extra precision) mode using the default parameters in the Glide program in accordance with our previously published protocol [16]. The obtained best docking scores or binding energy for EAI001 were estimated to be −11.472 kcal/mol, −11.532 kcal/mol, and −11.005 kcal/mol (Table 1), respectively. As shown in Figure 3, the docking conformation with the best docking score overlap well with original crystal conformation at the allosteric site of EGFR and the associated root mean-square deviations (RMSDs) were 0.048 Å, 0.178 Å, and 0.163 Å, respectively [16]. In general, when the RMSD is below 2.0 Å, molecular docking is reliable [17]. Our results indicated that the current docking simulation methods are suitable for the virtual screening of novel inhibitors. As a result, structure-based virtual screening was conducted by employing the Glide HTVS, SP and XP methods in this study.

### 2.2. Structure-Based Virtual Screening

Two databases, Chemdiv and Enamine, were rationally selected according to the structural diversity and commercial availability of the compounds. First, ChemDiv and Enamine databases were downloaded from the ZINC website [18,19], prepared using the Ligprep module, and filtered using Lipinski’s rule of five to improve compound drug-likeness [20]. The optimized databases were then submitted to the successive virtual screening workflow (VSW) using Glide 7.4 in Schrödinger suite 2017-1. Briefly, compounds were sequentially filtered using the HTVS, SP, and XP docking protocol, with the retaining ratios set as 1%, 10%, and 50%, respectively. At the final stage of VS, we conducted a visual inspection and selected compounds with key ligand–protein interactions necessary for the inhibition reported by us [16,21]. The ZINC code and the docking scores (G-score, Kcal/mol) based on XP mode are listed in Table 2, and the structures are shown in Appendix A. Generally, the lower the Glide score, the better the predicted affinity of the compounds binding to the protein target. The chosen compounds showed the best docking scores and interacted with several key kinase residues in EGFR allosteric site. Take ZINC49691377 as an example, the XP docking score was estimated to be −14.03 kcal/mol, which was lower than that of EAI045 and EAI001. ZINC49691377 showed necessary ligand–protein interactions for the inhibition (Figure 4). Specifically, 3-hydroxy-pyridine moiety stacks with the side chain of Leu 788, Ile 789, and the gatekeeper Met790, the 4-fluoro-phenyl moiety interacts with Phe 856 side chain through π-π stacking interaction, while the quinoline moiety extends into a hydrophobic pocket composed of Leu 747, Ile 759, Met 766 Leu 777, and Leu 788. One hydrogen bond was formed between N-atom and Asp 855 in the DFG motif located. On the other hand, a salt bridge was formed between 3-hydroxy-pyridine moiety and the catalytic residue Lys 745.

### 2.3. ADME and Toxicity Prediction

For the chosen 23 compounds, the drug-likeness and properties of absorption, distribution, and metabolism, excretion, and toxicity (ADMET) were predicted using QikProp. The results are shown in Table 2. Drug-likeness of compounds was assessed by observing the physicochemical properties like molecular weight (MW), predicted octanol/water partition coefficient (logP), predicted aqueous solubility (logS, S in mol/L), polar surface area (PSA). MW refers to the mass of a molecule, which evaluates drug-likeness or determine if a chemical compound with a certain pharmacological or biological activity has properties that would make it a likely orally active drug in humans. The logP value represents the partition coefficient between octanol and water, which is critical for measuring hydrophobicity of molecules and helps to evaluate absorption and distribution of drugs within the body. Aqueous solubility indicates the extent to which a molecule is soluble in water. The molecular polar surface area (PSA) is a physical chemical property describing the polarity of molecules. It is defined as the surface sum over all polar atoms, primarily oxygen and nitrogen, also including their attached hydrogens [22]. PSA is a descriptor that was appeared to relate well with passive molecular transport through membranes and, consequently, permits forecast of transport properties of drugs in the intestines and blood–brain barrier crossing. Molecules with a polar surface area of greater than 140 angstroms squared tend to be poor at permeating cell membranes, while a PSA less than 60 angstroms squared usually be good at permeating cell membranes. The ADMET profile includes Caco-2 permeability, MDCK permeability, blood–Brain barrier (BBB), and the blockage of HERG K^+^ channels was calculated. Caco-2 cells are a model for the intestine-blood barrier, and Caco-2 permeability is great markers of drug absorbance in the intestine [23]. Madin-Darby canine kidney (MDCK) monolayers, are widely used to make oral absorption estimates, the reason being that these cells also express transporter proteins, but only express very low levels of metabolizing enzymes [24]. They are also used as an additional criterion to predict BBB penetration. Thus, our calculated apparent MDCK cell permeability could be considered to be a good mimic for the BBB (for non-active transport). The blood/brain partition coefficients (logB/B), a predictor for access to the central nervous system (CNS), were computed, which can be used to predict their neurotoxicity. The human ether-a-go-go related gene (HERG) K^+^ channel appears to be the molecular target responsible for the cardiac toxicity of a wide range of therapeutic drugs [25]. HERG K^+^ channel blockers are potentially toxic and the predicted IC_50_ values often provide reasonable predictions for cardiac toxicity of drugs in the early stages of drug discovery [26]. In this work, the estimated or predicted IC_50_ values for blockage of this channel have been used to model the process. The recommended range for predicted log IC_50_ values for blockage of HERG K^+^ channels (logHERG) is >−5. Also, compounds that pass Lipinski’s rule exhibit good oral absorption in humans. Violations from Lipinski’s rule of five were also studied [27]. Compounds with fewer (or preferably no violations) of the rule are most probably orally available. The study of in silico ADMET predictions helps in the drug development process to produce novel molecules that are safe for human consumption and exhibit good oral absorption in humans. All 23 compounds displayed the ADMET properties within a range appropriate for human usage and showed no violations of Lipinski’s rule of five, showing their potential as drug-like molecules.

### 2.4. MD Simulations Analysis

To monitor the stability of the docked complex, molecular dynamic (MD) simulations were carried out using the Desmond 4.9 [28] in Schrödinger suite 2017-1. MD simulation of the docked complex of ZINC49691377-EGFR with the best docking score was performed for a total of 100 ns. For comparation, the co-crystal structure of EAI001 with EGFR was also subjected to 100 ns molecular dynamics.

In order to monitor the conformational stability throughout simulations, the root mean square deviation (RMSD) values of EGFR Cα atoms and the ligand were determined relative to the starting structure. The results were displayed in Figure 5 and Appendix A. RMSDs of carbon alpha atoms less than 3.00 Å during the simulation indicated that protein fluctuation was within acceptable variation, demonstrating the stability of the protein structure. The RMSDs of the ligand revealed that ZINC49691377 and EAI001 remained stable within the active pocket of EGFR during the simulation.

In order to examine the flexibilities of the individual residues that may have contributed to the overall fluctuations in the system, root mean square fluctuation (RMSF) of the complex was calculated (Figure 6 and Appendix A). Higher fluctuation values in the residues suggest greater flexibility in the course of MD simulations. In general, residues without any interactions are more flexible. As shown in Figure 6 and Appendix A, the two systems showed similar RMSF distributions and exhibited no unusual fluctuations. High fluctuations were observed in regions containing residue numbers 161–178 (Ala 859-Val 876). The RMSF profile was corroborated with the findings in our previous publication [16,21].

The most stable possible interactions between ligands and proteins over the MD simulation are better understood due to the visualizations of the protein–ligand contacts (Figure 7 and Appendix A). The main types of interactions that assisted in stabilizing the ligands were hydrogen bonding, hydrophobic, and water bridge interactions. For compound EAI001, Lys 745 formed a hydrogen bond with the of isoindolin-1-one moiety with a high percentage of 53%, aminothiazole moiety exhibited strong hydrophobic interactions with EGFR, and the phenyl substituent is in contact with Phe856. For ZINC49691377, Lys 745 directly formed two H-bond interactions with the hydroxyls on the pyridine ring and quinoline ring with high percentages of nearly 95% and 92%, respectively, and Asp 855 formed a hydrogen bond with ZINC49691377 with a percentage of nearly 99% over the course of MD simulation. In addition, ZINC49691377 formed hydrogen bonds with Thr 854 and Asp 855 via water bridge. This suggests that hydrogen bonding plays a significant role in accommodating and stabilizing ZINC49691377 at the binding site. In addition, hydrophobic interactions with Phe 723, Leu 747, Met 766, Leu 777, Leu 788, Met 790, Phe 856, and Leu 858 were demonstrated to be crucial for ligand binding. As illustrated in Figure 8 and Appendix A, the timeline representation of the interactions and contacts between ZINC49691377, EAI001 and EGFR during 100 ns MD simulation revealed that Lys 745 and Asp 855 were crucial for ligand binding within the active site.

The stability of the complex was also confirmed by measuring the ligand properties of the RMSD, rGyr (radius of gyration), intraHB (intramolecular hydrogen bond), MolSA (molecular surface area molecular surface area), SASA (solvent-accessible surface area), and PSA (polar surface area) in the EGFR protein pocket (Figure 9 and Appendix A), which kept an equilibrium throughout the 100 ns MD simulation.

### 2.5. The Anti-Proliferation Activity of the Selected Compounds

All 23 selected compounds were purchased from the commercial source for biological evaluation. The vendors had confirmed the purity of compounds by using LC-MS or NMR experiments. The anti-proliferation activity of these 23 compounds against NSCLC cell lines A549 harboring wide type EGFR, PC-9 expressing exon 19 deletion mutant, and H1975 expressing EGFR^L858R/T790M^ were assessed using MTT method, with EAI045 as positive control (Table 3). As shown in Table 3, EAI045 inhibited anti-proliferation against H1975 cells (EGFR ^L858R/T790M^) with IC_50_ value of 23.64 μM, and showed no obvious anti-proliferation against A549 (EGFR^WT^) and PC-9 (EGFR^Del 19^) cells. To our delight, four compounds (ZINC49691377, ZINC03876430, ZINC01201194, and ZINC13351329) demonstrated encouraging anti-proliferation activities. For example, ZINC01201194 displayed good anti-proliferative activity against PC-9 and H1975 with IC_50_ values of 2.30 μM, and 4.79 μM (Table 3), respectively. ZINC49691377 demonstrated good anti-proliferative activity against H1975 and PC-9 cells with IC_50_ values of 10.02 μM and 20.48 μM (Table 3 and Figure 10), respectively, which was more potent than the positive compound EAI045. We also determined the antiproliferative activity of ZINC49691377 against A549 cells, human epithelial cell line HaCaT and H3122 NSCLC cell line harboring the EML4-ALK fusion gene variant 1 (Figure 10). As shown in Figure 10, ZINC49691377 did not show obvious inhibitory activity against A549, HaCaT and H3122 cells (IC_50_ > 100 μM), showing good cellular selectivity.

### 2.6. The Cell Apoptosis of Compound ZINC49691377

To characterize the effect of ZINC49691377 on apoptosis progression, the Annexin-V/propidium iodide (PI) biparametric cytofluorometric assay was carried out in A549 (EGFR^WT^), PC-9 (EGFR^Del19^) and H1975 (EGFR^L858R/T790M^) cells (Figure 11). As shown in Figure 11, after treatment with ZINC49691377 at 2.5 μM, 5 μM and 10 μM, the percentage of late apoptotic cells in H1975 cell line increased to 10.6%, 14.5%, and 20.7%, and the percentage of early apoptotic cells increased to 5.74%, 11.5%, and 20.4%, respectively. And in PC-9 cell line, the percentage of late apoptotic cells increased to 11.0%, 17.2%, and 21.4%. The results stated that compound ZINC49691377 can effectively induce apoptosis of H1975 and PC-9 cells in a dose-dependent manner. However, ZINC49691377 had no significant effect on apoptosis of A549 cell line at given concentrations.

## 3. Materials and Methods

### 3.1. Protein Preparation

The three-dimensional atomic coordinates of EGFR bound with EAI001 (PDB ID 5D41) was extracted from the Protein Databank Bank (PDB) (http://www.rcsb.org/ (accessed on 8 June 2016)). Protein preparation was accomplished using the Protein Preparation Wizard in Maestro according to our previous publication [21]. The initial structure was preprocessed by assigning the corrected bond orders, adding the hydrogen atoms, filling in missing side chains and missing loops using Prime, and generating the protonation states of the ionizable residues using Epik at pH = 7. Then, the protein structure was minimized using OPLS3 force field [29] with the default cutoff RMSD value of 0.3 Å. The native ligand EAI001 was extracted from the crystal structure and prepared using the LigPrep module in Maestro 11.1.

### 3.2. Preparation of the Databases

Commercially available compounds of the ChemDiv (about 1,400,000 compounds) and Enamine (about 1,800,000 compounds) databases were downloaded from the ZINC website. Ligprep of the Schrödinger Suite was used to prepare the ligand. The possible stereoisomers, tautomers and ionization states were produced with Epik 3.9 at physiological pH. Then, the compounds were energy minimization using OPLS3 force field. The database was filtered using Lipinski’s rule of five [20] (specifically, a molecular weight (MW) less than 500, an octanol-water partition coefficient logPo/w no greater than 5, number of hydrogen bond donors no more than 5, number of hydrogen bond acceptors no more than 10). Then, the database was subjected to a subsequent virtual screening.

### 3.3. Docking-Based Virtual Screening

The prepared co-crystallized protein structure was employed for virtual screening based on molecular docking. Before proceeding to the docking procedure, the protein receptor grid defining the position and size of the active site for docking was generated by selecting the center of co-crystalized ligand EAI001 as the centroid of the grid box using a receptor grid generation tool with default settings. The co-crystalized ligand was redocked using the Glide HTVS, SP and XP with default parameters. The best Glide scored docking poses were chosen as the most likely binding conformation. Low RMSD values were achieved between the co-crystallized and docked pose, demonstrating that Glide HTVS, SP and XP were able to well reproduce the experimental conformation.

Therefore, we started the subsequent virtual screening using Glide HTVS, SP and XP with the same validation parameters. The top 1% of compounds from the initial HTVS docking were kept and rescored in SP mode. The chosen top 10% of compounds were subsequently subjected to extra precision (XP) docking analysis. After that, the top 50% of compounds were virtually analyzed and checked the essential interactions responsible for EGFR inhibition. Finally, 23 compounds were chosen based on the good XP GScore and key interactions with the residues of binding pocket.

### 3.4. Prediction of ADMET Properties

ADMET properties were predicted using the default mode of QikProp 5.1 implemented in the Schrödinger Suite. Properties assessed involved logPo/w (predicted octanol/water partition coefficient), logS (Predicted aqueous solubility), PSA (polar surface area), PCaco (Predicted apparent Caco-2 cell permeability), PMDCK (Predicted apparent MDCK cell permeability), logBB (the predicted partition coefficient of the brain/blood barrier), logHERG (Predicted blockage of the HERG K^+^ channels). And the violations of Lipinski’s rule of five (Ro5) were also calculated to evaluate drug-likeness.

### 3.5. Molecular Dynamics Simulations

Classical MD simulations were carried out using the Desmond program [28] in Schrödinger suite 2017-1. The simulated systems were created by the System Builder panel of Desmond by adding six Na^+^ to neutralize the system and setting the salt concentration to 0.15 M NaCl. The system was then dissolved in a 10 Å × 10 Å × 10 Å orthorhombic box TIP3P water model [30] with periodic boundary conditions. The energy of the solvated systems was first minimized in NVT ensemble for 20 ps at low temperature (10 K) to remove high energies in the predicted model. Then a seven-step equilibration of the systems was performed in NVT and NPT ensembles using the Desmond relaxation protocol. Finally, the 100 ns production simulations were carried out in the NPT ensemble at 303.15 K and standard pressure (1.01325 bar) with a time step of 2 fs [31]. Martyna-Tuckerman-Klein chain coupling scheme [32] and the Nosé–Hoover chain coupling scheme [33] were used to control pressure and temperature, respectively. A RESPA (reversible reference system propagator algorithms) integrator [34] was used to calculated nonbonded forces, with the short-range Coulomb interactions being created using a radius of cutoff 9 Å and long-range interactions being calculated using the smooth Particle Mesh Ewald. The SID (Simulation Interaction Diagram) program of Schrödinger suite was used to analyze trajectory.

### 3.6. Cancer Cell Proliferation Inhibition Assay

The anti-proliferative activities against human NSCLC-derived A549 cells (WT EGFR), PC-9 cells (746–750 deletion in exon 19 of EGFR), H1975 cells (L858R/T790M double mutant EGFR), HaCaT cells (immortalized human epithelial cell line), and H3122 cells (EML4-ALK rearrangement driven NSCLC cells) were determined using the standard MTT assay as reported previously [35], with EAI045 as the positive control. All cell lines were purchased from Shanghai Cell Bank, China Academy of Science (Shanghai, China). All chemicals and regents were bought commercially from Thermo Fisher Scientific (Waltham, MA, USA) or Sigma-Aldrich Chemical Company (St. Louis, MO, USA) and used without any additional purification. Briefly, cancer cells were seeded in 96-well plates at a density of 3–4 × 10^3^ cells per well (100 μL) and cultured at 37 °C in a humid environment with 5% CO_2_ for 24 h. On the second day, different concentrations of the test compounds were added to the fresh culture medium and the mixture was incubated for 72 h. After that, the tumor cells were then treated with the freshly produced MTT and incubated at 37 °C for 4 h. The formed formazan crystals in each well were dissolved in 100 μL DMSO. The absorbency was measured at 570 nm using a M1000Pro ELISA plate reader (Tecan, Männedorf, Switzerland). The half-inhibitory concentration (IC_50_) was estimated using non-linear regression analysis.

### 3.7. Cell Apoptosis Assay

Apoptosis assay was performed as previously described [35]. H1975 cells were seeded in 6-well plates at 3 × 10^5^ per well, and PC-9 cells and A549 were seeded at 2 × 10^5^ per well. One day later, different concentrations of compounds were added to continue incubation for 48 h. The original medium was collected, washed with pre-cooled PBS, pancreatin without EDTA were added, digestion was terminated with the collected original medium, the collected cells were centrifuged at 1000× *g*, 5 min, the cells were cleaned with PBS, and were centrifuged again. Then disposed with an Annexin V-FITC apoptosis detection kit (Beyotime-C1062L, BD Biosciences, San Jose, CA, USA) according to the instructions and apoptosis rate was determined by Flow Cytometry (LSRFortessa X-20, BD Biosciences, USA).

## 4. Conclusions

In this study, virtual screening based on molecular docking was employed to discover novel EGFR inhibitors, which was conducted on ChemDiv and Enamine commercial databases using Glide HTVS, SP and XP mode. After multi-step VS and visual evaluation, a total of 23 structurally unique and diverse compounds were chosen. ADMET properties of all compounds predicted by using Qikprop fall within an acceptable range. The molecular dynamics simulations (100 ns) demonstrated that ZINC49691377 formed a stable complex with EGFR and exhibited the conserved hydrogen bond interactions with Lys 745 and Asp855 of EGFR over the course of simulation. All compounds were further subjected into experimental testing. The representative ZINC49691377 showed good anti-proliferative activity against H1975 and PC-9 cells with IC_50_ values of 10.02 μM and 20.48 μM, respectively, with no obvious inhibitory activity against A549, HaCaT and H3122 cells (IC_50_ > 100 μM), showing good cellular selectivity. Meanwhile, apoptosis analysis indicated that compound ZINC49691377 can effectively induce apoptosis of H1975 and PC-9 cells in a dose-dependent manner, with no significant effect on the apoptosis of A549 cell line at given concentrations. These results indicate that compound ZINC4961377 with a new scaffold can be considered as a good point for further optimization.

## Figures and Tables

**Figure 1 ijms-25-01887-f001:**
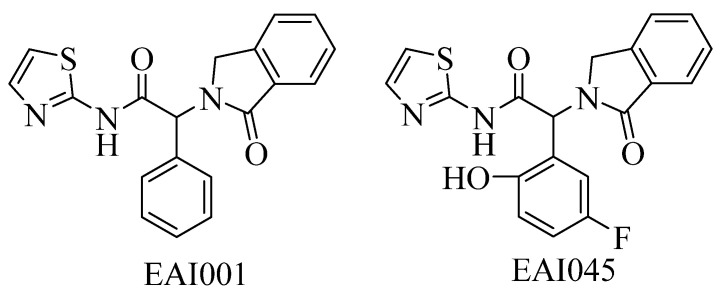
EGFR allosteric inhibitors EAI001 and EAI045.

**Figure 2 ijms-25-01887-f002:**
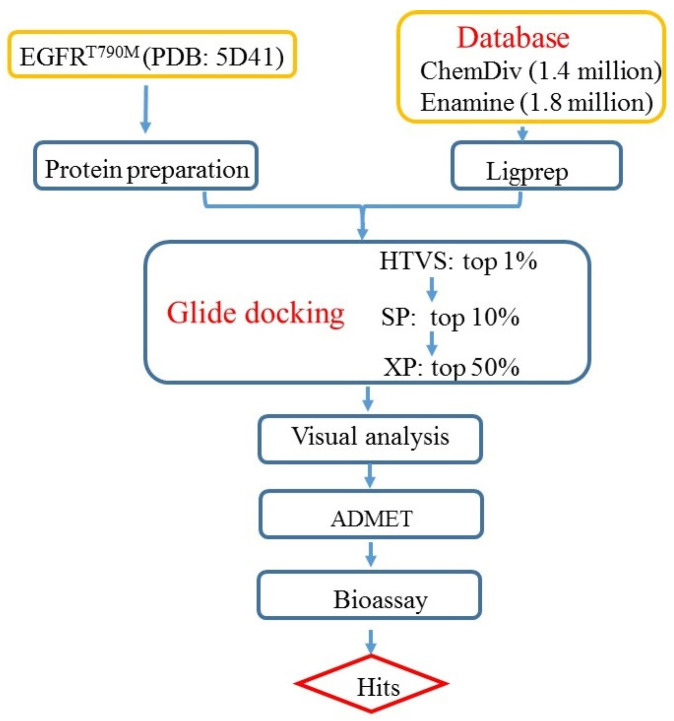
The screening flowchart of active molecules against EGFR.

**Figure 3 ijms-25-01887-f003:**
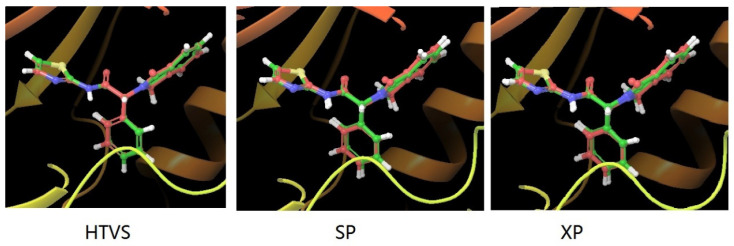
Structural comparison between co-crystallized conformation and re-docked conformation of EAI001 generated by Glide HTVS, SP and XP. Magenta and green represent co-crystallized and re-docked conformation, respectively.

**Figure 4 ijms-25-01887-f004:**
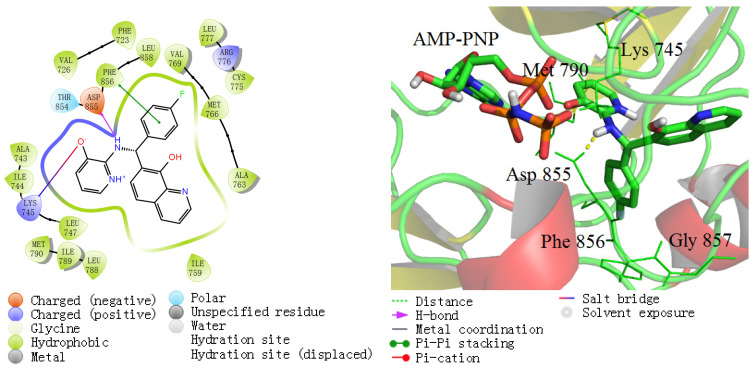
Binding interaction of the representative ZINC49691377 with EGFR.

**Figure 5 ijms-25-01887-f005:**
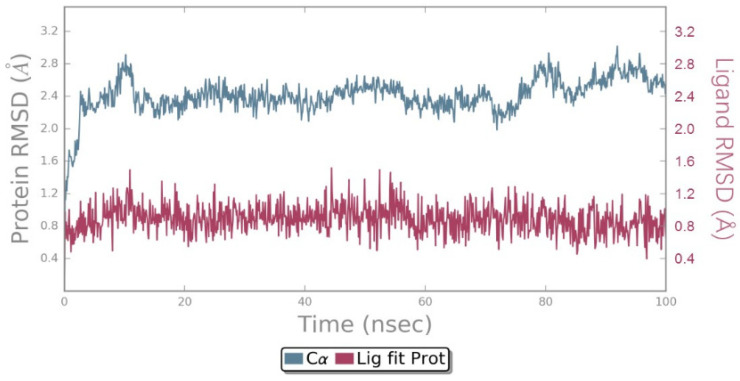
RMSDs of carbon alpha atoms (Å) and ligand atoms (Å) for the docked complex of ZINC49691377-EGFR during 100 ns MD simulation.

**Figure 6 ijms-25-01887-f006:**
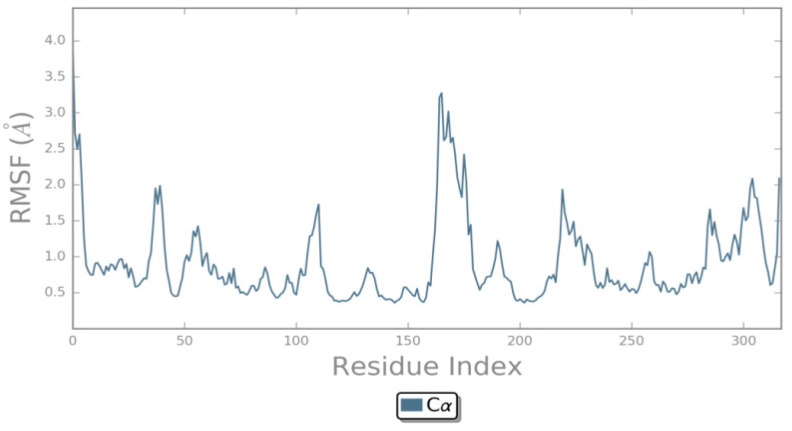
Root mean square fluctuation (RMSF, Å) of carbon alpha for the docked complex of ZINC49691377-EGFR during 100 ns MD simulation.

**Figure 7 ijms-25-01887-f007:**
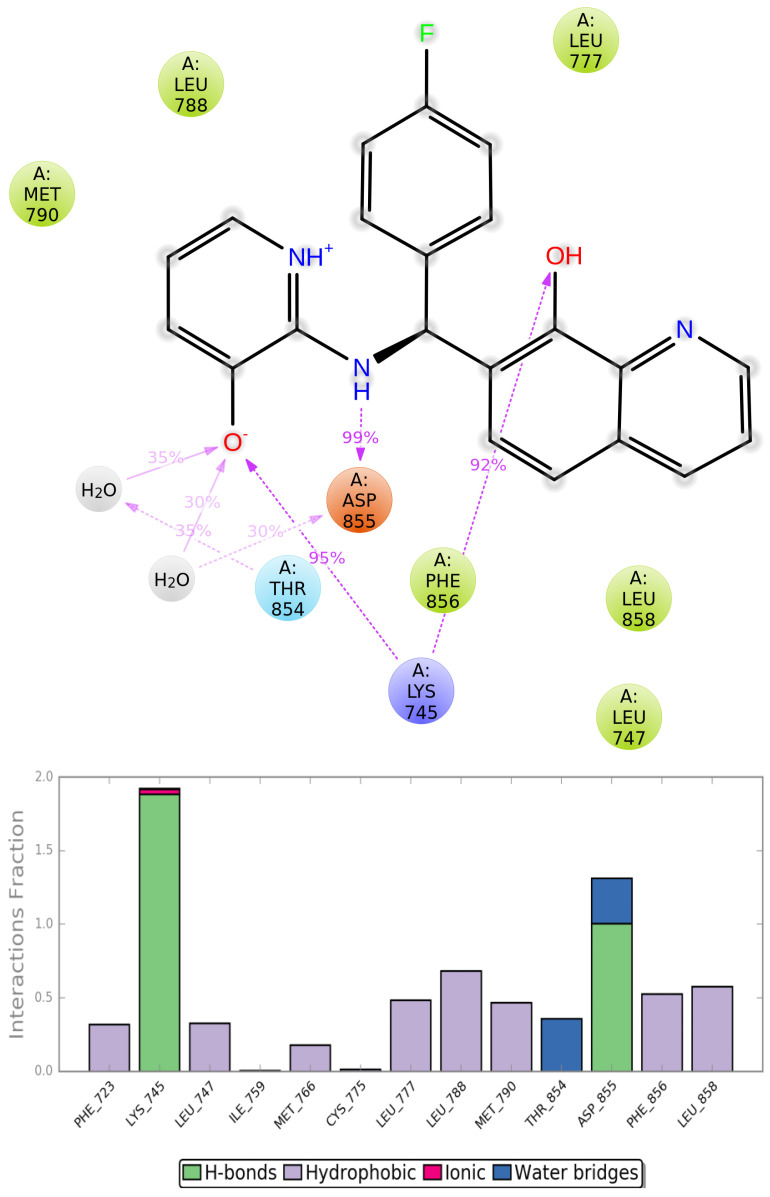
The 2D-interaction diagrams (**top**) and interaction histograms (**bottom**) of the simulated complex of ZINC49691377-EGFR during 100 ns MD simulation. Green, purple, red, and blue represent hydrogen bond interactions, hydrophobic interactions, ionic interactions, and water bridges, respectively.

**Figure 8 ijms-25-01887-f008:**
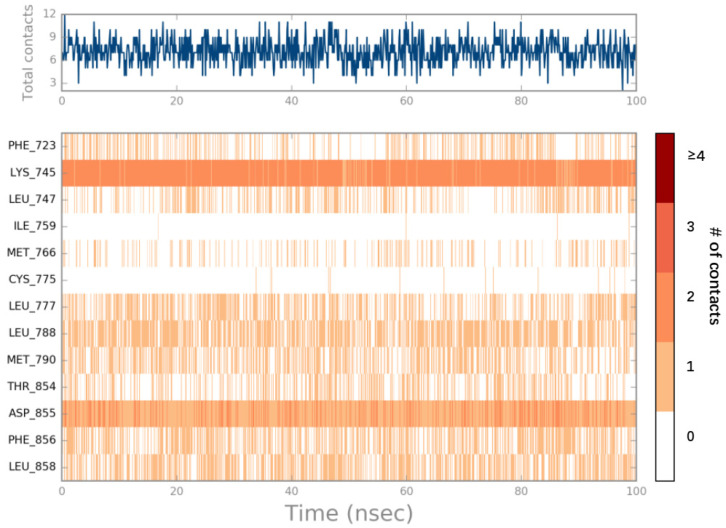
A timeline representation of the interactions and contacts between ZINC49691377 and EGFR during 100 ns MD simulation. The top panel shows the total number of specific contacts the protein makes with the ligand over the course of the trajectory. The bottom panel shows which residues interact with the ligand in each trajectory frame. Some residues make more than one specific contact with the ligand, which is represented by a darker shade of orange, according to the scale to the right of the plot.

**Figure 9 ijms-25-01887-f009:**
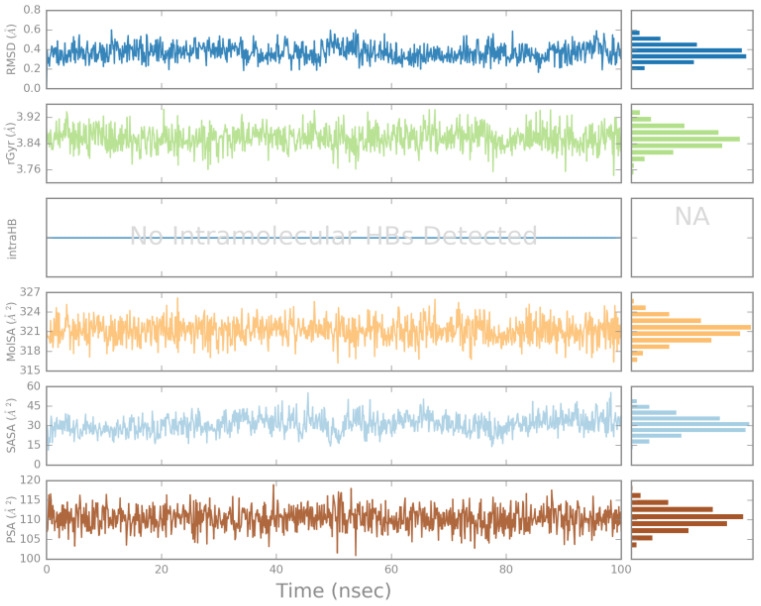
Ligand properties of ZINC49691377 in complex with EGFR throughout 100 ns MD simulation. RMSD: Root mean square deviation of a ligand with respect to the reference conformation. Radius of Gyration (rGyr): Measures the ‘extendedness’ of a ligand, and is equivalent to its principal moment of inertia. Intramolecular Hydrogen Bonds (intraHB): Number of internal hydrogen bonds (HB) within a ligand molecule. Molecular Surface Area (MolSA): Molecular surface calculation with 1.4 Å probe radius. This value is equivalent to a van der Waals surface area. Solvent Accessible Surface Area (SASA): Surface area of a molecule accessible by a water molecule. Polar Surface Area (PSA): Solvent accessible surface area in a molecule contributed only by oxygen and nitrogen atoms.

**Figure 10 ijms-25-01887-f010:**
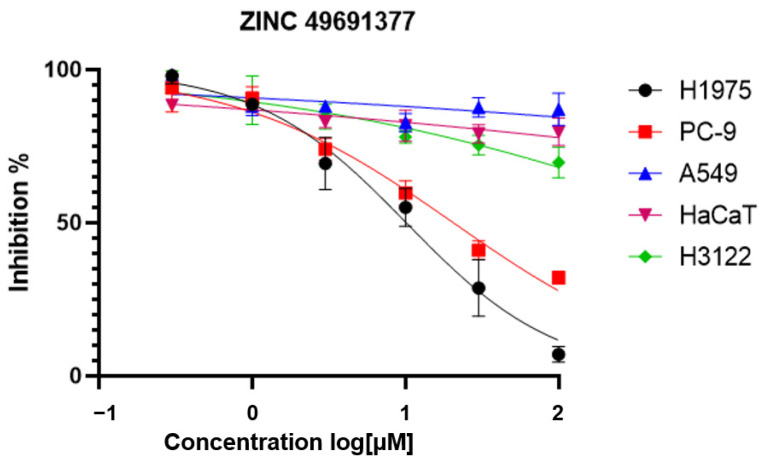
The dose–response curves of the representative compound ZINC49691377.

**Figure 11 ijms-25-01887-f011:**
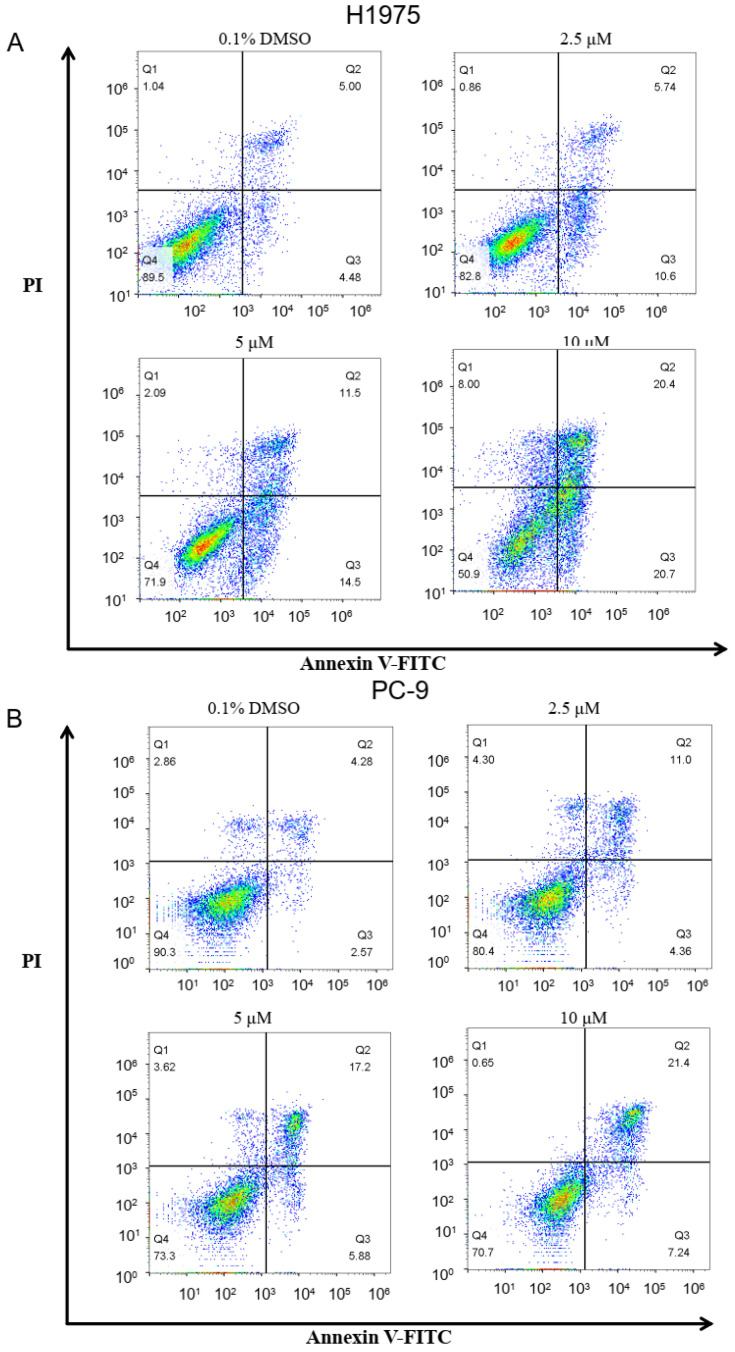
Apoptosis induction of compound ZINC49691377 in A549, PC-9 and H1975 cells. (**A**–**C**) Flow cytometry analysis of Annexin V-FITC Apoptosis Detection Kit of apoptotic cells following 0.1% DMSO, 2.5 μM, 5 μM, and 10 μM of ZINC49691377 for 48 h. (**D**–**F**) Percentage of live, early apoptosis, late apoptosis, and dead cells. The experiments were carried out twice, independently.

**Table 1 ijms-25-01887-t001:** The docking results of co-crystallized EAI001 generated by Glide HTVS, SP and XP.

	HTVS	SP	XP
G-score (Kcal/mol)	−11.472	−11.532	−11.005
RMSD (Å)	0.048	0.178	0.163

**Table 2 ijms-25-01887-t002:** ZINC code, Glide XP score (G-Score) value, ADMET properties of selected compounds.

		G-Score (kcal/mol)	MW ^a^	logPo/w ^b^	logS ^c^	PSA ^d^	PCaco ^e^	PMDCK ^f^	logBB ^g^	logHERG ^h^	Violation of Ro5 ^i^
	EAI045	−11.96	383.38	2.93	−4.67	86.8	877.17	1538.77	−0.60	−6.32	0
1	ZINC49691377	−14.03	361.38	4.14	−5.06	71.00	1072.92	966.13	−0.67	−6.56	0
2	ZINC09616958	−13.03	356.42	4.60	−5.38	69.31	907.73	445.56	−0.93	−6.68	0
3	ZINC09775243	−12.33	361.83	4.98	−5.49	48.91	2785.90	3693.51	−0.11	−6.37	0
4	ZINC10910059	−12.19	355.40	3.70	−4.70	74.03	1303.15	658.64	−0.63	−6.73	0
5	ZINC89827617	−12.07	329.79	3.47	−4.33	91.10	259.21	543.26	−0.80	−4.02	0
6	ZINC22017635	−11.99	342.40	4.95	−5.86	53.87	2283.11	1946.38	−0.18	−6.47	0
7	ZINC53674458	−11.93	328.37	4.37	−5.43	55.48	2342.50	1241.45	−0.35	−7.23	0
8	ZINC69419433	−11.89	348.40	3.66	−5.49	96.22	758.32	366.85	−0.97	−6.61	0
9	ZINC15778674	−11.53	363.35	4.10	−5.50	82.31	1408.00	1056.08	−0.52	−7.12	0
10	ZINC29507326	−11.52	365.38	3.72	−5.02	66.11	1657.03	1656.87	−0.20	−6.36	0
11	ZINC05577262	−11.52	363.35	3.80	−5.18	78.59	1483.36	1371.80	−0.35	−6.52	0
12	ZINC03876430	−11.42	341.41	4.58	−6.18	58.56	2121.51	1115.36	−0.27	−6.47	0
13	ZINC18205922	−11.41	347.80	4.92	−5.84	47.30	3375.47	4550.21	0.01	−6.88	0
14	ZINC00036286	−11.36	343.42	4.85	−5.17	48.57	3155.77	1713.26	−0.22	−6.19	0
15	ZINC20531081	−11.31	342.40	4.21	−5.18	61.33	2484.77	1323.15	−0.30	−7.00	0
16	ZINC89756684	−11.30	331.38	4.22	−5.47	65.15	2241.61	1183.76	−0.37	−6.89	0
17	ZINC01201194	−11.28	382.25	4.87	−6.31	57.21	2291.39	6343.36	0.11	−6.39	0
18	ZINC01108543	−11.27	368.43	4.49	−5.03	56.21	1887.49	982.99	−0.27	−5.35	0
19	ZINC13351329	−11.08	355.44	4.77	−5.87	54.80	1767.73	1368.58	−0.17	−5.27	0
20	ZINC04918676	−11.06	356.42	3.55	−4.44	70.31	1237.96	623.09	−0.45	−5.44	0
21	ZINC04784223	−11.06	371.44	4.75	−5.43	70.92	1544.54	791.45	−0.65	−6.62	0
22	ZINC43232082	−10.99	374.36	4.35	−5.30	62.00	2051.67	5859.35	−0.05	−4.64	0
23	ZINC00178936	−10.70	345.42	4.33	−5.97	56.46	2085.50	2130.51	−0.23	−6.85	0

^a^ MW: Molecular weight of the molecule (Range or recommended values: 130.0–725.0). ^b^ logPo/w: Predicted octanol/water partition co-effcient log P (acceptable range: −2.0 to 6.5). ^c^ logS: Predicted aqueous solubility; S in mol/L (acceptable range: −6.5 to 0.5). ^d^ PSA: Polar surface area (acceptable range: 7.0 to 200.0). ^e^ logHERG: K^+^ ion channel related toxicity (logHERG: concern below −5). ^f^ PCaco: Predicted apparent Caco-2 cell permeability in nm/s. (Range or recommended values <25 poor, >500 great); ^g^ PMDCK: Predicted apparent MDCK cell permeability in nm/s (acceptable range: >500 is high, <25 is poor). ^h^ QPlogBB: the predicted partition coefficient of the brain/blood barrier (logBB: −3.0 to 1.2). ^i^ Violation of RO5: number of violations of Lipinski’s rule five, consisting of five rules for drug-like compounds, which require MW < 500, logPo/w < 5, number of donor H-bond ≤ 5, number of acceptor H-bond ≤ 10).

**Table 3 ijms-25-01887-t003:** The antiproliferative activity of selected compounds.

	Title	Antiproliferative Activity (IC_50_, μM) ^a^		
		A549	PC-9	H1975
	EAI045	>100	>100	23.64 ± 4.78
1	ZINC49691377	>100	20.48 ± 0.03	10.02 ± 0.02
2	ZINC09616958	32.32 ± 1.57	82.34 ± 2.20	40.36 ± 0.95
3	ZINC09775243	96.17 ± 2.04	82.68 ± 2.47	61.10 ± 6.04
4	ZINC10910059	19.03 ± 0.76	13.98 ± 0.18	15.06 ± 0.50
5	ZINC89827617	>100	>100	>100
6	ZINC22017635	39.33 ± 3.95	18.95 ± 3.60	13.60 ± 1.46
7	ZINC53674458	>100	90.13 ± 1.75	78.74 ± 8.80
8	ZINC69419433	>100	>100	14.21 ± 4.57
9	ZINC15778674	>100	67.20 ± 2.20	40.13 ± 4.50
10	ZINC29507326	>100	85.25 ± 19.00	43.91 ± 15.16
11	ZINC05577262	>100	>100	74.26 ± 4.82
12	ZINC03876430	34.53 ± 0.74	38.05 ± 7.36	9.69 ± 0.91
13	ZINC18205922	12.52 ± 0.58	33.98 ± 4.50	20.24 ± 0.43
14	ZINC00036286	87.23 ± 0.68	70.19 ± 3.65	75.78 ± 4.24
15	ZINC20531081	>100	93.65 ± 6.67	62.98 ± 7.65
16	ZINC89756684	73.01 ± 10.27	73.06 ± 7.68	43.12 ± 1.34
17	ZINC01201194	8.20 ± 1.18	2.30 ± 0.63	4.79 ± 1.11
18	ZINC01108543	37.23 ± 3.20	30.95 ± 0.55	38.13 ± 1.22
19	ZINC13351329	10.80 ± 0.94	6.00 ± 0.76	16.52 ± 2.55
20	ZINC04918676	>100	>100	>100
21	ZINC04784223	>100	46.42 ± 1.97	>100
22	ZINC43232082	28.04 ± 2.02	30.69 ± 0.14	23.27 ± 1.62
23	ZINC00178936	>100	>100	>100

^a^ The values are the average of two independent experiments performed in duplicate.

## Data Availability

Data is contained within the article and Appendix A.

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
