# Peer review of "Identification of New EGFR Inhibitors by Structure-Based Virtual Screening and Biological Evaluation"

_ijms, 2024, doi:10.3390/ijms25031887_

Round 1

Reviewer 1 Report

Comments and Suggestions for Authors

Based on virtual screening authors determine that compound ZINC49691377 not only meets the criteria for drug-likeness and ADMET , but also forms a stable complex with EGFR T790 mut during 100 ns molecular simulation. Anti-EGFR proliferation activity in NSCLC cell lines could also demonstrate selectivity of the compound for mutant cells vs EGFR-wt cells. Based on this, the authors define  ZINC49691377  as a molecular scaffold for further development of novel EGFR inhibitors.

Here are some points that could improve the quality of the manuscript:

-  Figure 4 needs to be improved ( e.g. crossing of the arrows, color of the arrows)

- dose-response curves need to be presented in addition to table 3. at least for few mutant vs wt cells.

- authors do not elaborate the parameters they used for the structrual based screen, they also do not explain why they decided to use the selected databases.

more information on the setting of the database required

- based on the results shown, authors overinterpret the data.

- pharmacokinetics and dynamics are missing.

- are 100 ns enough?

- In line 157 they argue they used 23 compounds however, the table 2 (line 160) shows 24 compounds that have been used for Glide XP scoring.

- discussion needs to be improved

future aspects/limitations need to be discussed.

Comments on the Quality of English Language

The manuscript is well-written. No major issues

Author Response

-  Figure 4 needs to be improved ( e.g. crossing of the arrows, color of the arrows)

A: Thank you very much for your suggestion. Figure 4 has been improved. There are no arrows crossing in the picture.

- dose-response curves need to be presented in addition to table 3. at least for few mutant vs wt cells.

A: Thank you very much for your suggestion. The dose-response curves of the represent compound ZINC49691377 has been presented in the manuscript. The revisions and changes were highlighted in Yellow.

Figure 10. The dose-response curves of the represent compound ZINC49691377 .

- authors do not elaborate the parameters they used for the structrual based screen, they also do not explain why they decided to use the selected databases.

more information on the setting of the database required

A: Thank you very much for your suggestion. During the virtual screening, PDB ID: 5D41 was selected, which has a resolution of 2.31 Å. Two databases, Chemdiv and Enamine, were rationally selected according to the structural diversity and commercial availability of the compounds. First, ChemDiv and Enamine databases were downloaded from the ZINC website, prepared using the Ligprep module, and filtered using Lipinski’s rule of five to improve compound drug-likeness. The optimized databases were then submitted to the virtual screening workflow (VSW) using Glide software. The revisions and changes were highlighted in Yellow.

- based on the results shown, authors overinterpret the data.

A: Thank you very much for your suggestion. We have Modified some results.

 The revisions and changes were highlighted in Yellow.

- pharmacokinetics and dynamics are missing.

A: Thank you very much for your suggestion. Among them, the most promising hit ZINC49691377 demonstrated excellent anti-proliferation activity against H1975 and PC-9cells, with no significant anti-proliferation activity against A549, HaCaT and H3122 cells. Based on virtual screening and bioassays, ZINC4961377 can be considered as an excellent starting point for the development of new EGFR inhibitors. In the future, we will optimize the structure of the compound to improve the activity and selectivity, so there have been no pharmacokinetics and dynamics research on the hit compound.

- are 100 ns enough?

A: Thank you very much for your suggestion. As displayed in Fig. 5, the RMSDs of carbon alpha atoms and ligand atoms for the docked complex of ZINC49691377-EGFR were less than 3.00 Å and no largescale conformational changes were observed, which indicated that the system was stable during the 100 ns MD simulation

- In line 157 they argue they used 23 compounds however, the table 2 (line 160) shows 24 compounds that have been used for Glide XP scoring.

A: Thank you very much for your suggestion. The first compound is the compound EAI045, which was listed for comparation. For clarity, we have modified the table.

The revisions and changes were highlighted in Yellow.

- discussion needs to be improved

future aspects/limitations need to be discussed.

A: Thank you very much for your suggestion. We have improved the discussion, and discussed future aspects/limitations.

The revisions and changes were highlighted in Yellow.

Reviewer 2 Report

Comments and Suggestions for Authors

The manuscript "Identification of New EGFR inhibitors by structure-based virtual screening and biological evaluation" is devoted to computer analysis of molecules of epidermal EGFR inhibitors.

This study is very relevant in terms of developing drugs for the treatment of non-small cell lung cancer. Today this is a very relevant socially significant disease, which is often incurable due to the very small number of drugs approved for use in medicine.

In this work, good and high-quality studies of synthesized molecules of epidermal growth factor inhibitors were carried out using computer analysis methods. The cytotoxicity of the compounds was tested using the standard MMT test.

It is unfortunate that the authors were unable to present studies of epidermal growth factor inhibition using various protein assay methods. The reviewer hopes that in subsequent works the authors will take this request into account.

The MTT test is not always an indicative study, and cells may die from a general toxic effect, and not from the effects of inhibition of epidermal growth factor specifically. It is also necessary to include in the article studies on the induction of apoptosis in A549 cells, which may indirectly indicate the antitumor potential of these synthesized compounds.

The article may be published in this journal.

Author Response

A: Thank you very much for your suggestion. We have added the apoptosis of compound ZINC49691377 in A549, PC-9, and H1975 cells.

2.6. The Cell Apoptosis of Compound ZINC49691377

To characterize the effect of ZINC49691377 on apoptosis progression, the Annexin-V/propidium iodide (PI) biparametric cytofluorometric assay was carried out in A549 (EGFRWT), PC-9 (EGFRDel19) and H1975 (EGFRL858R/T790M) cells. As shown in Fig. 11, After treatment with ZINC49691377 at 2.5 μM, 5 μM and 10 μM, the percentage of late apoptotic cells in H1975 cell line increased to 10.6%, 14.5%, and 20.7%, and the percentage of early apoptotic cells increased to 5.74%, 11.5%, and 20.4%, respectively. In PC-9 cell line, the percentage of late apoptotic cells increased to 11.0%, 17.2%, and 21.4%. The results stated that compound ZINC49691377 can effectively induce apoptosis of H1975 and PC-9 cells in a dose-dependent manner. However, ZINC49691377 had no significant effect on apoptosis of A549 cell line at given concentrations.

Figure 10. Apoptosis induction of compound ZINC49691377 in A549, PC-9 and H1975 cells. (A.C.E).Flow cytometry analysis of Annexin V-FITC Apoptosis Detection Kit of apoptotic cells following 0.1% DMSO, 2.5 μM, 5 μM, and 10μM of ZINC49691377 for 48 h. (B.D.F) Percentage of live, early apoptosis, late apoptosis, and dead H1975 cells. The experiments were carried out twice independently.

The revisions and changes were highlighted in Yellow.

Reviewer 3 Report

Comments and Suggestions for Authors

The manuscript sounds interesting even if sometimes poorly detailed or motivated. Please, revise the manuscript bearing in mind the following suggestions:

-          Introduction:

-          Figure 3 should be added at the end of the introduction section as a workflow of the whole study.

-          The main findings and the future prosecution of this work should be described at the end of the introduction.

Results and discussion

-          2.1 validation of docking; please revise this title as molecular docking protocol assessment

-          Lines 67, please add the proper reference for the cited PDB code

-          Control compounds for the development/discussion of the molecular docking studies and MD simulations should be mentioned and compared with those of the newly selected compounds.

-          Control compounds for the biological assays should be included and discussed.

-          Please, move all the figures after the first sentence in which they have been mentioned.

Author Response

  Figure 3 should be added at the end of the introduction section as a workflow of the whole study.

A: Thank you very much for your suggestion. Figure 3 has been moved to the end of the introduction section as a workflow of the whole study.

-          The main findings and the future prosecution of this work should be described at the end of the introduction.

A: Thank you very much for your suggestion. The main findings and the future prosecution of this work has been added at the end of the introduction.

The revisions and changes were highlighted in Yellow.

Results and discussion

-          2.1 validation of docking; please revise this title as molecular docking protocol assessment

A: Thank you very much for your suggestion. The title has been changed to molecular docking protocol assessment.

The revisions and changes were highlighted in Yellow.

-          Lines 67, please add the proper reference for the cited PDB code

A: Thank you very much for your suggestion. The reference has been added.

The revisions and changes were highlighted in Yellow.

-          Control compounds for the development/discussion of the molecular docking studies and MD simulations should be mentioned and compared with those of the newly selected compounds.

A: Thank you very much for your suggestion. The co-crystal structure of EAI001 with EGFR was also subject to 100ns molecular dynamics. The results were provided in Figure S1-S5 in supporting information. In the manuscript, the results were compared with ZINC49691377.

The revisions and changes were highlighted in Yellow.

-          Control compounds for the biological assays should be included and discussed.

A: Thank you very much for your suggestion. The biological assays of control compound (EAI045) was listed in TABLE 3. In the manuscript, the results were also discussed and compared with zinc49691377.

EAI045 exhibited anti-proliferation activity against H1975 cells (EGFR L858R/T790M) with IC50 value of 23.64 μM, and showed no obvious anti-proliferation activity against A549 (EGFRWT) and PC-9 cells (EGFRDel 19). ZINC49691377 demonstrated good anti-proliferative activity against H1975 and PC-9 cells with IC50 values of 10.02 μM and 20.48μM (Table 3 and Figure 10), respectively, which was more potent than the positive compound EAI045.

The revisions and changes were highlighted in Yellow.

-          Please, move all the figures after the first sentence in which they have been mentioned.

A: Thank you very much for your suggestion. All the figures have been moved after the first sentence in which they have been mentioned.

The revisions and changes were highlighted in Yellow.

Round 2

Reviewer 2 Report

Comments and Suggestions for Authors

I am completely satisfied with the article as presented.

Reviewer 3 Report

Comments and Suggestions for Authors

The manuscript has been revised according to most of the proposed revisions.